# Feeding Cows with Olive Cake Enriched in Polyphenols Improves the Sustainability and Enhances the Nutritional and Organoleptic Features of Fresh Caciocavallo Cheese

**DOI:** 10.3390/foods13203320

**Published:** 2024-10-18

**Authors:** Federica Litrenta, Fabrizio Cincotta, Nunziatina Russo, Carmelo Cavallo, Cinzia Caggia, Annalisa Amato, Vincenzo Lopreiato, Maria Merlino, Antonella Verzera, Cinzia Lucia Randazzo, Luigi Liotta

**Affiliations:** 1Department of Biomedical, and Dental Sciences and of Morphological and Functional Imagines (Biomorf), University of Messina, 13 G. Palatucci Street, 98168 Messina, Italy; federica.litrenta@unime.it; 2Department of Veterinary Sciences, University of Messina, 13 G. Palatucci Street, 98168 Messina, Italy; fabrizio.cincotta@unime.it (F.C.); carmelo.cavallo@unime.it (C.C.); annalisa.amato@unime.it (A.A.); vincenzo.lopreiato@unime.it (V.L.); maria.merlino@unime.it (M.M.); antonella.verzera@unime.it (A.V.); 3Department of Agriculture, Food and Environment (Di3A), University of Catania, 100 Santa Sofia Street, 95123 Catania, Italy; nunziatinarusso83@gmail.com (N.R.); ccaggia@unict.it (C.C.)

**Keywords:** cheese, fatty acids, polyphenols, sustainability, consumer health

## Abstract

In this study, Caciocavallo, a typical cheese produced in Sicily Island (Italy), was obtained from the milk of dairy cows fed with and without enriched olive cake (ECO and CTR, respectively) in order to evaluate nutritional, microbiological, volatile, and sensory differences in cheeses. ECO cheese showed greater (*p* < 0.05) MUFA and PUFA and polyphenols content and lower SFA content than CTR cheese. Microbiological analyses revealed the absence of Salmonella, *Listeria monocytogenes*, *Escherichia coli*, and *E. coli* O157, and no significant differences in the viable counts of the remaining microbial groups analyzed, between samples. Thermophilic lactococci were more prevalent in ECO cheese. The implementation of a culture-independent method, such as PCR-DGGE analyses, revealed the presence of a more diverse microbial population in both cheeses. Regarding the volatile compounds profile, long-chain free fatty acids were more abundant in the ECO cheese, resulting in a healthier free fatty acid profile. This study also showed that, especially for their appearance and taste, consumers mostly appreciated the ECO cheese. The results show that using enriched olive cake could enhance the sustainability and the quality of Ragusano cheese, improving not only the health of its consumers but also positively influencing tastes and acceptability.

## 1. Introduction

The production of European cheeses made from bovine milk has shown, in recent years, an increase of +5.8% from 2018 (9,070,916 tons) to 2023 (9,614,010 tons) [1], with a growth forecast in 2024, arousing particular interest in the environmental impact that the cheese production industry may have worldwide. In a recent study on a life-cycle assessment of cheese production [2], it emerged that raw milk production was the major environmental impact category. Currently, there are several studies on the use of waste by-products as natural biosources of beneficial added-value compounds [3], which both decrease the environmental impact caused by food waste and, at the same time, enhance food nutritional quality [4]. To achieve these goals, different strategies are focused on the integration of agro-industrial byproducts in animal nutrition, due to their considerable amounts of bioactive compounds [5]. There are several studies on the effect of the integration of byproducts into the diet of dairy cows [6], such as dried apple pomace [7], pelleted citrus pulp [8], the mix of tomato and apple pomace [9], and olive cake (OC) [10,11]. These studies evidence how the use of by-products can give added value to the final product: for example, Amato et al. [12] have recently observed a significant increase in monounsaturated fatty acid (MUFA) and polyunsaturated fatty acid (PUFA) and a decrease in saturated fatty acid (SFA) in the milk of cows fed with olive cake enriched with polyphenols. Furthermore, the addition of olive polyphenols in the diet may modify the microbiological profile of cheese as reported by Calabrese et al. [13], which, after olive cake supplementation in dairy cows’ diet, showed an improvement in the microbiological quality of cheese.

Caciocavallo Ragusano is a traditional cheese produced in Sicily, with a production in 2023 of 199 tons and an export amounting to 1,637,550 kg [1], made with a traditional manufacturing process (e.g., using traditional wood tools), where the fermentation is driven by indigenous LAB, which arises from raw milk, a biofilm present on the inner surface of the vats, and the environment [14]. In this context, fed supplementation with olive cake could represent a promising strategy to improve both the sustainability of Ragusano cheesemaking and quality of the traditional cheese, making it more enjoyable for consumers. Thus, this study aimed to evaluate the nutritional value, microbiological profiles, sensory features, and volatile aroma compounds of Caciocavallo Ragusano produced with milk obtained from cows fed with olive cake enriched in polyphenols (ECO). 

## 2. Materials and Methods

### 2.1. Animal Management and Treatment

The experimental protocol was approved by the Ethical Committee of the Department of Veterinary Science, University of Messina, Italy (code 041/2020). The research complied with guidelines of Good Clinical Practices (The European Agency for the Evaluation of Medicinal Products [15]) and the Italian and European regulations on animal welfare (Council of The European Union [16]).

The study was conducted on a commercial dairy farm in Ragusa (Sicily, Italy), located 500 m above sea level, with 460 healthy multiparous Friesian dairy cows enrolled in the trial. The animals were randomly allocated into two homogeneous groups (n = 230/group) according to body condition score (BCS; 2.43 ± 0.26), days in milk (DIM; 113 ± 47 d), and milk yield (MY; 31.42 ± 3.28 kg/d). All the animals were housed in the same free stall barn and managed in accordance with local traditional practices, with deep straw bedding. All the animals had access to grass pasture for a minimum of 6 h during daylight (0800 to 1400 h). Cows were fed with 2 meals of forage (grass hay), offered at intervals of 12 h, and 2 meals of concentrates (isoenergetic and isoproteic), individually delivered during milking time (0630 and 1830 h). All animals had free access to drinking water. To cover energy and protein requirements, for 2.5 kg of milk produced, 1 kg of concentrate was delivered. Thus, on average, animals were fed daily 10.23 ± 2.11 kg of concentrate as dry matter (DM). All cows were milked twice daily (0630 and 1830 h). The experimental group (ECO) received a concentrate supplemented with 7% unconventional olive cake (OC) on a dry matter (DM) basis, and the control group (CTR) received a concentrate without olive cake supplementation. The ingredients and chemical composition of the concentrates have been published elsewhere [12] and are reported in Appendix A. The enriched OC with high polyphenols content was supplemented according to approved UE disciplinary “QS Sicilia”, which aims to recover agro-industrial by-products. The enriched OC was obtained by a 2-stage oil mill, where food-grade hypocloridric acid 25% E 507 for stabilization and wastewater from oil production were added, with the aim of increasing the polyphenols content; a flowchart of the enriched OC production is presented in Figure 1.

Ingredients and chemical composition [17] of concentrates, enriched olive cake, pasture, and grass hay are reported in Appendix A.

### 2.2. Cheese Making

After 3 weeks from the start of the feeding treatment (April 2023), the milk from the two groups was collected separately and transported in refrigerated tanks (4 ± 2 °C) to commercial cheesemaking located in Ragusa. Four hundred (400) liters of milk from each group were used to produce Caciocavallo Ragusano cheese (Appendix A), according to the traditional flowchart reported by Licitra et al. [18]. Samples were brined for 10 days, vacuum-packed, and transported to the Departments of Veterinary Sciences of the University of Messina, Italy, and to the Department of Agriculture, Food and Environment, University of Catania, Italy, for further analyses.

### 2.3. Chemical, Polyphenols, and Fatty Acid Analyses

The physico-chemical composition of milk samples (pH, SH°, fat, total protein and casein, lactose, and total solids) was determined using a pH meter (Hannah HI9023 pH meter, Hannah Instruments, Keysborough, VIC, Australia) and a Milkoscan FT3 (Foss Electric, Hilleroed, Denmark) apparatus.

Moisture content (AOAC, 2000; method 948.12 [19]), protein content (AOAC 2000; method 920.123 [20]), and fat content (AOAC 2000; 98 method 933.05 [21]) were measured on cheese samples, in triplicate. Fatty acid methyl esters (FAMEs) were obtained from milk and cheese fat trans-methylated and analyzed through GC-FID, as described by Attard et al. [10]. Individual FAMEs were identified, based both on retention time and on comparison with a standard mixture of 37 pure components (Supelco 37 Component FAME Mix, Merck Life Science, Milano, S.r.l, Italy). Single standards (marked with an asterisk) not included in the mix were identified as a percentage of the area and the retention time of the individual standard. Total polyphenols of milk and cheese were measured according to the Folin–Ciocalteu method, as described by Singleton et al. [22]. To analyze the total phenolic content in cheese extract using the Folin–Ciocalteu method, 1.5 mL of ultrapure water was added to 300 µL of cheese extract. Afterward, 2.5 mL of Folin–Ciocalteu reagent and then 2 mL of Na_2_CO_3_ were added. The tubes were then incubated in the dark for 90 min at room temperature, and the absorbance was measured at 725 nm against the blank solution. The standard used for calculation was a gallic acid solution.

### 2.4. Microbiological Analyses

Milk samples were analyzed for the presence of *Enterobacteriaceae*, *Escherichia coli* and coliforms, total mesophilic bacteria, *Listeria monocytogenes*, and *Salmonella*, as previously reported by Calabrese and coworkers [13]. In addition, lactic acid bacteria (LAB) were counted on de Man, Rogosa, and Sharpe (MRS) agar, and enterococci on kanamycin azide agar, both incubated anaerobically at 37 °C for 48 h, whereas Sabouraud Dextrose Agar (SDA), with chloramphenicol, was incubated at 25 °C for 72 h, and then the counting of yeasts was performed.

Outer and inner samples of cheese (i.e., rind and core, respectively; weight 500 g) were collected using aseptic techniques with a sterile knife from 0-day-ripened cheese wheels. Twenty-five grams of cheese samples were homogenized with 225 mL of sterile peptone water (Oxoid, Basingstoke, UK) in the Stomacher (Interscience) for 2 min at 260 rpm. Serial tenfold dilutions were prepared, and 100 μL of each dilution was inoculated in duplicate as previously reported by Calabrese et al. [13]. All the media were from Liofilchem (Roseto degli Abruzzi, Italy).

### 2.5. Total DNA Extraction and PCR-DGGE Analyses

The CTR and ECO milk and cheese samples were subjected to direct extraction of total bacterial DNA as previously reported. After DNA extraction, PCR amplification was performed according to the protocols reported, using the universal PCR primers U968-GC and L1401-r, targeting the V6 to V8 regions of eubacterial 16S rDNA [23]. After verification by electrophoresis on a 1.2% (*w*/*v*) agarose gel, 20 µL of the PCR products were used for Denaturing Gradient Gel Electrophoresis (DGGE) analysis on the Dcode System apparatus (BioRad, Hercules, CA, USA). The run was carried out in 8% polyacrylamide (acrylamide/bis-acrylamide mix 37.5:1, *w*/*v*) gels with a 40.0 to 60.0% urea–formamide (*w*/*v*) gradient (100% denaturant was 7 M urea plus 40%, *w*/*v*, formamide) increasing in the direction of electrophoresis. Gels were subjected to a constant voltage of 85 V for 8 h at 60 °C for 16 h. The DNA bands were visualized by silver staining and were developed as previously described [24].

### 2.6. Volatile Aroma Compound Analysis

The volatile aroma compounds of milk, feeds, and cheese were analyzed by head space–solid-phase micro-extraction–gas chromatography–mass spectrometry (HS-SPME-GC-MS). An amount of 20 mL of milk, 2 g of feed + 15 mL of saturated NaCl, and 10 g of finely cut cheese + 10 mL of saturated NaCl solution were placed into a 40 mL glass vial and equilibrated for 20 min at 40 °C, respectively. Then, a triphasic DVB/CARB/PDMS fiber was exposed for 30 min in the headspace for the volatile compounds extraction and successively placed for 3 min in the injection port of the GC kept at 260 °C. A Shimadzu GC 2010 Plus gas chromatographer interfaced with a TQMS 8040 triple quadrupole mass spectrometer (Shimadzu, Milan, Italy) was used for the GC-MS analyses with the following conditions: capillary column, VF-WAXms (60 m. 0.25 mm i.d.; coating thickness 0.25 μm); injector temperature, 260 °C; injection mode, splitless; oven temperature, 45 °C held for 5 min, then increased to 80 °C at a rate of 10 °C minute^−1^, and to 240 °C at a rate of 2 °C minute^−1^; carrier gas, helium; flow, 1 mL min^−1^; transfer line temperature, 250 °C; acquisition range 40–400 m z^−1^; scan speed, 1250 amu s^−1^. Compound identification was undertaken using mass spectral data, NIST’20 (NIST/EPA/NIH Mass Spectra Library, Wiley USA) FFNSC 3.0 database, linear retention indices (LRIs), literature data, and the injection of the available standards. The LRIs were calculated according to the van den Dool and Kratz equation. The results were expressed as peak area percentages.

### 2.7. Qualitative Descriptive Analysis

A qualitative descriptive analysis (QDA) of Caciocavallo Ragusano cheese was carried out according to Merlino et al. [25]. The panel was trained according to ISO 8586-1:2023;9 [26]; a common vocabulary was established for explaining the sensory descriptors and to familiarize the panel with scales and procedures. Each descriptor was described and explained to avoid any doubt about the relevant meaning. Based on the frequency of citations, 21 descriptors were selected (yellow color, color uniformity, presence of fracture, presence of lighter colored spots, glossy, milk odor, butter odor, green odor, hay odor, olive cake odor, ripened cheese odor, rancid odor, salty taste, bitter taste, sour taste, pungent taste, juiciness, tenderness, friability, springiness, greasiness, and adhesiveness). The sensory panel evaluated the intensity of the selected descriptors with a score included between 1 (absence of the sensation) and 9 (extremely intense). Each judge evaluated the samples in four sessions. All evaluations were carried out from 10.00 to 12.00 a.m. in individual booths illuminated by white light. The order of presentation was randomized among judges and sessions. Water and unsalted crackers were provided to judges between samples. All data were acquired by a direct computerized registration system (FIZZ Byosistemes. ver. 2.00 M, Couternon, France). The results were expressed as the average for each sensory attribute.

### 2.8. Consumer Acceptability Test

Consumer acceptability was evaluated by voluntary consumers randomly selected among the students and personnel of the University of Messina (n = 80, 37 males and 43 females, 24–60 years). Consumers evaluated 4 attributes, namely appearance, odor, taste, and texture, with a 9-point hedonic scale (9 = like extremely and 1 = dislike extremely).

### 2.9. Statistical Analysis

Statistical analyses were performed using SPSS 13.0 for Windows [27] (SPSS Inc., Chicago, IL, USA). The initial multivariate dataset comprised 6 samples of Caciocavallo cheese and included 54 variables. These data were categorized into two groups based on dietary conditions, specifically CTR and ECO. The significance of the differences between the groups was initially determined using the non-parametric Mann–Whitney U test. Following this, the data were normalized to ensure the independence of scaling factors across the various variables. The suitability of the initial data was assessed with the Kaiser–Meyer–Olkin (KMO) test and Bartlett’s test, which led to the application of principal component analysis (PCA) to differentiate samples according to their diet.

For the microbiological data, one-way analysis of variance (ANOVA) was conducted, followed by Tukey’s post hoc testing, using Statistica software (version 10.0 for Windows, TIBCO Software v. 13.5.0, Palo Alto, CA, USA) [28]. Statistical significance was accepted at *p* ≤ 0.05.

In the analysis of sensory characteristics and volatile components, XLStat software, version 2019.1.2 (Addinsoft, Damremont, Paris, France) was utilized. Two-way analysis of variance (ANOVA) and Duncan’s multiple range test were applied at a 95% confidence level to evaluate significant differences between the samples.

## 3. Results and Discussion

### 3.1. Chemical Composition

The chemical composition of milk and Caciocavallo Ragusano cheese is shown in Table 1 and in Table 2, respectively. No differences in the chemical composition of milk were detected between ECO and CTR. This result shows a different trend from that reported by Chiofalo et al. [29] and Castellani et al. [30], where the milk protein of animals fed diets supplemented with olive cake was higher than in the relevant control groups. However, differences in chemical composition were observed between ECO and CTR cheese. In fact, the percentage of total lipids of cheese was influenced by the dietary treatment (*p* ≤ 0.05). Cheeses made from milk in the CTR group had higher total lipids (27.49%) than cheese in the ECO group (25.03%). However, no significant differences (*p* > 0.05) were found in cheese protein between the groups.

The fatty acid profiles of milk and cheese are summarized in Table 1 and Table 2, respectively. The analysis reveals a consistent pattern in the fatty acid profiles of both milk and cheese. Notably, compared with CTR samples, the ECO samples exhibited an increase in monounsaturated fatty acids (MUFA) and polyunsaturated fatty acids (PUFA), alongside a decrease in saturated fatty acids (SFA). Specifically, there was an increase in fatty acids such as C16:1, C18:1 cis9, and C18:3 cis6 cis9 cis12, while a decrease was noted in C12:0, C14:0, C17:0, and C18:0. Among the MUFAs, the levels of C16:1, C18:1 cis9, and C18:1 trans9 showed statistically significant differences between diet groups (*p* ≤ 0.05), with C18:1 trans9 being notably higher in the CTR samples compared with ECO samples. Conversely, C18:3 cis6 cis9 cis12 was the only PUFA that demonstrated a significant difference, being higher in the ECO samples compared with CTR samples. In terms of fatty acid composition, the addition of olive cake led to a reduction in both short- and medium-chain fatty acids, which are synthesized entirely “de novo” or partially by the mammary gland using rumen-derived acetate and beta-hydroxybutyrate, respectively [31]. Medium-chain fatty acids can increase low-density lipoprotein (LDL) cholesterol levels in the blood if not balanced with appropriate linoleic acid levels [32]. The elevated C18:1 cis9 content in ECO cheeses likely stems from two factors: the high oleic acid content present in olive pomace and the potential conversion of stearic acid to oleic acid in the mammary gland facilitated by delta9-desaturase [33], which is driven by the stearoyl-coenzyme A desaturase gene [34], in accordance with Chiofalo et al. [29] and Vargàs-Bello-Pérez et al. [35] working with sheep, and with Terramoccia et al. [36] working in buffaloes supplemented with olive cake. Similar findings were reported by Tzamaloukas et al. [37] on milk samples from small ruminants fed with olive pomace, Symeou et al. [38] in studies with sheep, and Castellani et al. [30], Neofytou et al. [39], and Amato et al. [12] in studies with dairy cows fed with olive cake.

Olive cake supplementation resulted in an increase in total polyphenols in ECO samples of milk and cheese (Table 1 and Table 2), corroborating that polyphenols are absorbed in the gastrointestinal tract [40]. However, even in the CTR samples, there was a low content of polyphenols. However, the higher amount of polyphenols in ECO cheese and milk reflects the significant amounts of polyphenols present in the OC [41]. It has been well established that these bioactive compounds lead to beneficial effects for human health, as they reduce the formation of free radicals and harmful oxidative events in the metabolism [42].

### 3.2. Principal Component Analysis

The Kaiser–Meyer–Olkin sampling adequacy measure came out with a value of 0.66, and Bartlett’s sphericity test showed a chi-square value of 710.16; thus, the correlation matrix was factored and fit for PCA.

According to the Kaiser–Guttman criterion, three principal components were extracted (eigenvalues: 18.25, 2.02, and 1.31), explaining 93.81% of the total variance (79.33%, 8.78%, and 5.71%, respectively) for Caciocavallo cheese, whereas the three principal components extracted for milk (eigenvalues: 18.18, 2.63 and 1.03) explained 92.04% of the total variance (78.64%, 7.68% and 5.68%, respectively). Analysis of the correlation matrix showed that the highest positive correlations were observed among C14:0 lipids (0.982), MUFA-C22:0 (0.976), and MUFA-C18:1 cis9 and PUFA-C15:1 (0.960), whereas the highest negative correlations were observed for MUFA-C14:0 (−0.988), MUFA-C18:0 (−0.956), and C22:0 lipids (−0.949). Figure 2 shows 2D scatterplots on the plane defined by PC1 and PC2 for the Caciocavallo cheese and milk samples.

Two groups can be clearly distinguished for cheese and two groups for milk: the ECO samples are separated from the CTR samples on the first component, which explains 79.336% and 78.683% of the total variances; the ECO samples are positioned at positive values of PC1 and are characterized by higher values of MUFA, C18:2 cis9 cis12, C22:0 and polyphenols, while the CTR samples are positioned at negative values of PC1 and have higher values of C12:0, C14:0 and lipids.

### 3.3. Microbiological Analysis

Microbiological analyses revealed the compliance of milk quality with the legal requirements (Commission Regulation (EC) No 2073/2005 of 15 November 2005 on microbiological criteria for foodstuffs [43]) for the production of raw milk cheeses. Indeed, the *Salmonella* spp. and *L. monocytogenes* milk pathogens were absent, whereas the count of *S. aureus* was below the detection limits (Table 3).

Although many factors can influence the microbial composition of raw milk [44,45] such as milking conditions, temperature, duration of milk storage and transport as well as the diet of the cows [46,47,48] no statistical differences (*p* > 0.05) were observed for the analyzed microbial groups, except for *E. coli*, and thermophilic lactococci densities (Table 3). Indeed, compared to CTR milk, in ECO samples, the number of *E. coli* bacteria was lower, while the thermophilic lactococci were more abundant. Overall, both LAB and the total mesophilic bacteria were the most abundant counted groups, with a density above 4.00 log cfu/mL in both CTR and ECO milk, followed by lactococci and enterococci. Concerning *Enterobacteriaceae*, molds, and yeasts, the viable count was below 3.00 log cfu/mL and comparable for both milk samples, whereas the lowest count was observed for total coliforms, especially in ECO milk.

Regarding cheese samples, a general increase in almost all microbial groups investigated was observed. The levels of the viable count of the CTR and ECO cheese samples are shown in Table 4.

Overall, all undesired species such as *Salmonella* spp., *L. monocytogenes*, *E. coli*, and *E. coli O157*, generally associated with poor hygiene of dairy production, were never detected in the analyzed samples, reinforcing the safety of the dairy environment [14]. Moreover, the viable counts of the remaining analyzed microbial groups did not significantly differ (*p* > 0.05) among all samples, except for the thermophilic lactococci whose density was significantly higher (*p* ≤ 0.05) in ECO cheeses (Table 4). Although the cell density of thermophilic LAB usually increases after cheese manufacturing, and it is well known that the microbial dynamics leading from raw cows’ milk to pasta filata cheese include all typical manufacturing steps [49], the observed increase in thermophilic lactococci levels in milk and later in cheeses here detected can probably be explained by the resistance of this group to polyphenol content of the olive cake supplementation. A previous work revealed the high resistance of lactococci to different classes of polyphenols, intentionally inoculated into raw ewe’s milk, showing their ability to perform rapid acidification [50]. It is also noteworthy that Lactococcus genus significantly contributes to the texture and flavor characteristics of fermented products by enhancing the rheological properties of fermented milk products [51] through the production of extracellular polysaccharides. In this study, the lactococci counts were between 6.91 and 6.68 log cfu/g, and the presumptive LAB count ranged from 7.34 to 7.03 log cfu/g in the CTR sample and the ECO cheese, respectively. Notably, the pro-technological groups, including LAB and both mesophilic and thermophilic cocci, dominated the microbial community, which, together with enterococci, were not affected by the thermal treatment applied during stretching, whose consistent increase could be due to later milk contact with the wooden vat [52]. It is well known that these bacteria play a crucial role during cheese fermentation, mainly in the development of sensorial characteristics and, generally, in the consistent contribution to the typicality of traditional cheeses [53]. Although the stretching operation, typical of Caciocavallo Ragusano cheese making, is known to exert a sanitizing effect [54], the thermal shock applied with stretching did not determine the reduction in the presumptive coagulase-negative staphylococci group in cheeses, where their levels were quite consistent (4.48 in the CTR and 5.29 log cfu/g in the ECO samples). Concerning the enterococci group, the value reached was about 5.00 log cfu/g for both samples, while the Enterobacteriaceae count was between 2.54 and 2.27 log cfu/g in the CTR and ECO cheese, respectively. Finally, no marked differences were observed between the cheeses for both total mesophilic bacteria and eumycetic population, which, compared to milk, increased in the same way.

### 3.4. PCR-DGGE Analyses

The DGGE profiles of the milk and cheese samples are shown in Figure 3.

Overall, the analyses showed differences in terms of the number of bands or relative abundance and intensity between the two types of samples and within the same group. In particular, the DGGE profiles of the CTR and ECO milk samples (lines 1 and 3, respectively) consisted of a few bands located at different heights within the gel, showing substantial differences between the two samples. In particular, one to two marked bands were observed in each sample, while both profiles were characterized by a greater number of weak bands, occasionally found in the corresponding cheese samples. Similarly, the DGGE profiles of the cheese (lines 2 and 4) shared some bands, of varying intensity, one of which was also with the ECO milk sample, where it was more prominent. Furthermore, the ECO cheese showed several weaker bands, not revealed in the CTR sample, presumably attributable to other species involved in the cheesemaking. These results confirm the importance to coupled different techniques (culture-dependent and independent) to study in greater depth microbial populations and their dynamics during cheese production.

### 3.5. Volatile Constituents

#### 3.5.1. Feeds and Olive Cake

In Table 5, the volatile percentage composition in single compounds and classes of substances for the concentrates, CTR and ECO, and olive cake are reported.

As results from the table, a large number of volatiles were identified, belonging to different classes of substances such as aliphatic acids, alcohols, ketones, esters, and aldehydes; moreover, terpenes, lactones, furans, and aromatic compounds were identified too.

The olive cake used in this research was characterized by high amounts of aldehydes, which constituted about 44% of the volatile fraction, with nonanal the main compounds; free fatty acids (FFA) followed. The large number of aliphatic aldehydes agrees with the higher level of oleic acid in the olive oil; in fact, aldehydes are common products of the decomposition of hydroperoxides developed by the oxidation of unsaturated fatty acid.

Comparing the composition of the CTR concentrate with ECO, an increase in aldehyde compounds resulted in agreement with the olive cake composition; otherwise, a decrease in alcohols mainly due to 3-methyl-1-pentanol resulted in the ECO concentrate. A large amount of 3-methyl-1-pentanol was present in the CTR concentrate; it is a plant metabolite but could be due to *Saccharomyces cerevisiae* fermentation.

#### 3.5.2. Milk and Cheese

In Table 6, the volatile percentage composition in single compounds and classes of substances for the milk and cheese, both CTR and ECO, are reported. The volatiles belonged to different classes of substances, all these well known in the volatile fraction of dairy products, such as aliphatic acids, alcohols, ketones, esters, and aldehydes; moreover, terpenes, lactones, and aromatic compounds were identified too.

Aliphatic acids are the main classes of substances in the milk and cheese samples. The short-chain acids are responsible for the sour taste, while long-chain acids are odorless; the unsaturated ones such as (E)-2-Decenoic acid have a more intense odor. Short-chain free fatty acids (SCFFA) have strong sweaty, cheesy, and lipolysis notes, low flavor thresholds, and are considered key aroma compounds in dairy products. In particular, butanoic acid (cheesy, dairy, buttery) has been demonstrated to be the marker aroma compound of raw cow milk.

Contrastingly, medium-chain free fatty acids (MCFFA) such as C_10_ and C_12,_ for their high flavor thresholds, influence the dairy products’ aroma less. Comparing the acid amount in CTR and ECO milk, there was a higher amount in the ECO, which had the highest amount of long-chain free fatty acids (LCFFA), while the CTR had the highest amounts of medium- and short-chain ones. Thus, the ECO cheese showed a healthier FFA profile.

The amount of FFA in the cheese samples reflects the milk composition with the highest amount in the ECO and a different ratio between short-, medium-, and long-chain fatty acid amounts.

The differences in the volatile FFA here reported are in agreement with Calabrese et al. [13], who demonstrated an increase in unsaturated volatile FFA in cheese when olive cake was added to the feed. Of interest is the difference in the ketone amount and, in particular, that of 2-ketones. Their amount was lower in the ECO cheese. They originate from the β-oxidation of fatty acids to β-ketoacids and decarboxylation to alkan-2-ones with one less C-atom; these are finally transformed in secondary alcohols, with a reversible reaction in aerobic conditions. A similar behavior was shown by acetoin and diacetyl (2,3 butanedione), with the lowest amount in the ECO cheese. Moreover, the ECO cheese was characterized by greater number of alcohols, mainly ethanol and isoamyl alcohol; the methyl-branched alcohols may derive from the reduction of aldehydes formed from amino acid via Strecker degradation. The terpene fraction seems to be less influenced by the addition of olive cake to the feed.

### 3.6. Sensory Analysis and Acceptability

Figure 4 reports the results of the QDA regarding the cheese CTR and ECO. 

Among the identified and considered descriptors, only a few significant statistical differences according to the fed composition were displayed. As regards appearance, ECO resulted in a more “yellow color” and in a lower “presence of lighter colored spots”; the “milk odor” and the “sour and piquant taste” intensity were higher in the CTR cheese, and the “sweet taste” in the ECO one. The differences in the color and appearance of the ECO samples could be due to the presence of carotenoids in the olive cake since it has been demonstrated to be an important source of these substances; as regards the CTR cheeses, the higher intensity of “milk odor” could be related to the amount of 2-ketones in the volatile fraction, while that of piquant and sour taste to the highest amount of SCFFA [55]. Figure 5 shows the results of the consumer’s acceptability test.

The consumers mostly appreciated the ECO cheese, especially for its appearance and taste. A more intense yellow color, together with a sweeter and less sour and piquant taste, is related to the consumer’s acceptability.

## 4. Conclusions

Local cheeses are characterized by unique flavors that reflect the environment in which the cheese is made, which includes the animal feed, the climate, the environment and the traditional processing method, which make the product unique. The production of a local cheese as Caciocavallo Ragusano, with the integration of olive cake enriched in polyphenols in the diet of dairy cows, can constitute a valid opportunity to nutritionally enrich the cheese thanks to the presence of polyphenols and unsaturated fatty acids, which are positively related to human health; moreover, the phenolic component of olive cake did not negatively affect the microbial composition of Caciocavallo Ragusano, which appears more diverse and with a prevalence of the thermophilic lactococci group. The volatile profile of the ECO cheese reflects the milk composition and showed a lower amount of 2-ketones arising from the oxidation of fatty acids and larger number of alcohols and LCFFA. Furthermore, this study showed a good acceptability of ECO cheese for consumers. With the growing awareness of consumer of health and quality, the acceptability of cheese provides an important economic impact for the local economy, since in a competitive market, consumer acceptability drives innovation. This study suggests that enriched olive cake boosts not only the sustainability but also the quality of Ragusano cheese.

## Figures and Tables

**Figure 1 foods-13-03320-f001:**
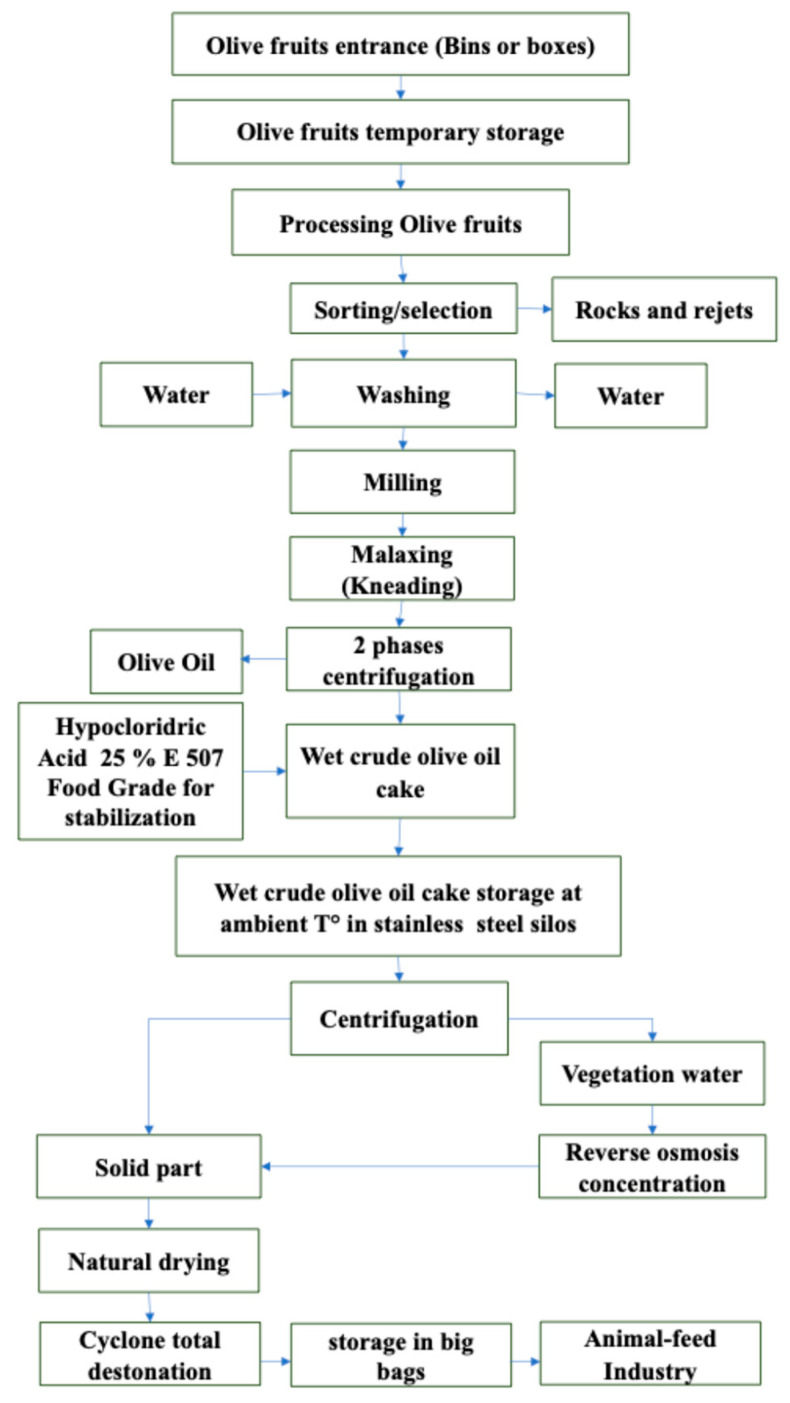
Destoned and enriched olive cake production (Patent N°: 001428707).

**Figure 2 foods-13-03320-f002:**
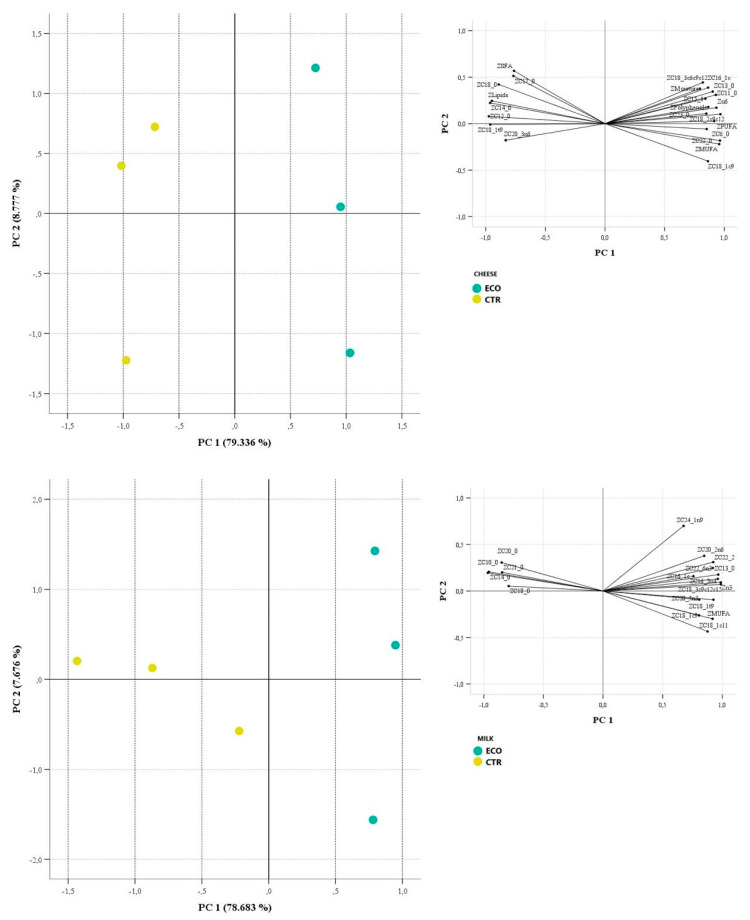
Two-D scatterplots for cheese and milk samples categorized by diet. Insert: loading plot for PC1 and PC2. CTR: milk and cheese made by dairy cows fed concentrate without olive cake supplements. ECO group: milk and cheese made by dairy cows fed concentrate with 7% enriched olive cake as 7% of a DM.

**Figure 3 foods-13-03320-f003:**
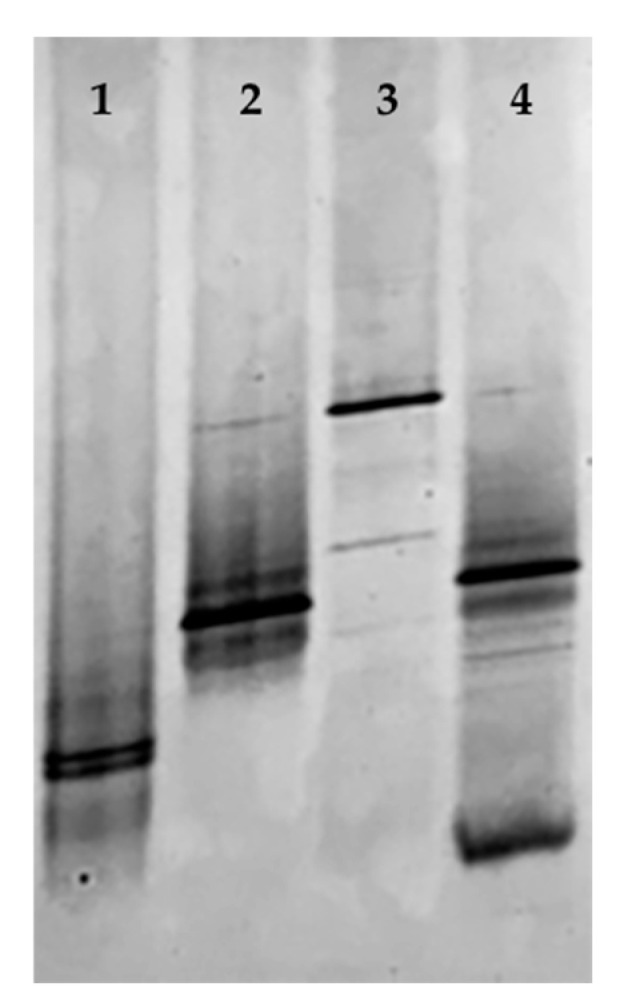
PCR-DGGE profiles of CTR milk (line 1), CTR cheese (line 2), ECO milk (line 3), and ECO cheese (line 4). CTR: milk and cheese made by dairy cows fed concentrate without olive cake supplements. ECO: milk and cheese made by dairy cows fed concentrate with 7% enriched olive cake as 7% of a DM.

**Figure 4 foods-13-03320-f004:**
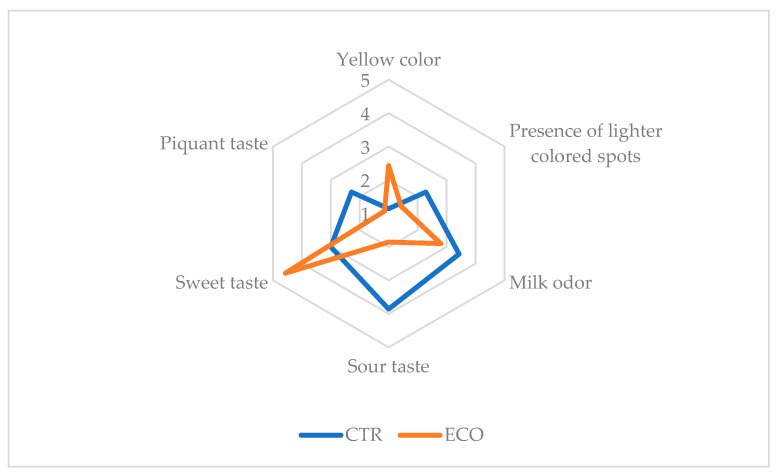
QDA spider plot of significant descriptors of Caciocavallo cheese. CTR: milk and cheese made by dairy cows fed concentrate without olive cake supplements. ECO: milk and cheese made by dairy cows fed concentrate with 7% enriched olive cake as 7% of a DM.

**Figure 5 foods-13-03320-f005:**
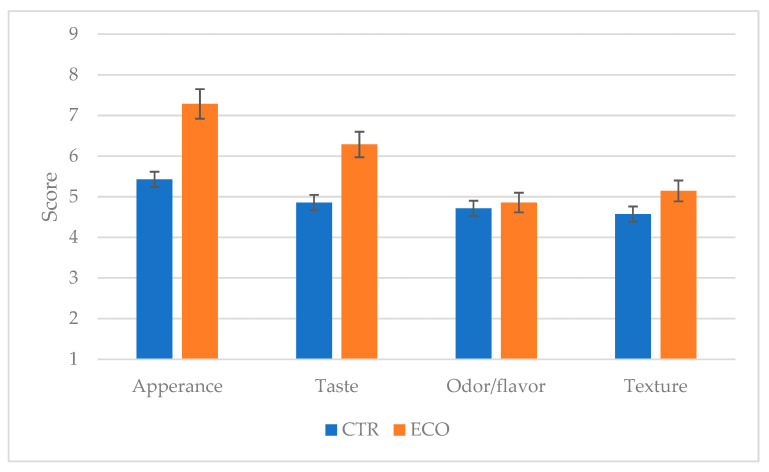
Consumer acceptability of Caciocavallo cheeses. 1: dislike extremely, 9: like extremely. CTR: milk and cheese made by dairy cows fed concentrate without olive cake supplements. ECO: milk and cheese made by dairy cows fed concentrate with 7% enriched olive cake as 7% of a DM.

**Table 1 foods-13-03320-t001:** Chemical composition, total polyphenols, and fatty acid profile of milk from cows of CTR group (without ECO) and cows of ECO group (with the integration of 7% of ECO in the concentrate).

Milk				
	Groups		SEM ^1^	*p*-Value
Item	CTR	ECO		
PH	6.68	6.65	0.07	0.26
SH°	3.2	3.2	0.01	0.31
Moisture (%)	86.82	88.21	0.09	0.51
Total proteins (%)	3.36	3.30	0.02	0.27
Casein (%)	2.64	2.59	0.02	0.21
Total lipids (%)	3.76	3.59	0.08	0.06
Lactose (%)	4.54	4.49	0.03	0.33
Total polyphenols (mg/kg)	62.13	109.25	0.12	0.04
Fatty acids (g/100 g FA)				
C4:0	2.47	2.51	0.13	0.70
C6:0	2.06	1.96	0.18	0.85
C8:0	0.97	1.02	0.05	0.26
C10:0	2.77	2.60	0.04	0.55
C11:0	0.06	0.13	0.45	0.05
C12:0	3.52	3.15	0.11	0.09
C13:0	0.09	0.13	0.07	0.04
C14:0	12.92	12.10	0.04	0.05
C15:0 iso *	0.13	0.14	0.07	0.19
C15:0 anteiso *	0.01	0.01	0.0001	1.00
C15:0	1.53	1.48	0.04	0.51
C16:0	28.14	27.02	0.02	0.06
C17:0	1.22	0.93	0.16	0.05
C18:0	13.50	12.17	0.06	0.05
C20:0	0.56	0.30	0.38	0.09
C21:0	0.07	0.05	0.19	0.06
C22:0	0.04	0.09	0.16	0.05
C23:0	0.08	0.09	0.09	0.05
C24:0	0.02	0.01	0.05	0.20
SFA	70.17	65.92	0.04	0.05
C14:1	0.91	0.95	0.06	0.50
C15:1	0.12	0.11	0.17	0.66
C16:1	1.79	1.93	0.05	0.05
C16:1 trans *	0.01	0.01	0.000	1.00
C16:1 n-5	0.01	0.01	0.01	1.00
C17:1	0.27	0.30	0.11	0.27
C18:1 cis9	15.91	18.26	0.08	0.05
C18:1 trans9	0.68	0.78	0.09	0.05
C18:1 cis11 *	3.08	3.64	0.11	0.07
C18:1 trans11 *	0.25	0.31	0.17	0.13
C20:1 n-11	0.06	0.06	0.08	0.98
C22:1 n-9	0.01	0.01	0.36	0.94
C24:1 n-9	0.03	0.04	0.05	0.11
MUFA	23.14	26.42	0.07	0.05
C16:3 n-4 *	0.04	0.07	0.30	0.06
C18:2 cis9 cis12	1.38	1.60	0.21	0.09
C18:2 trans9 trans12	0.67	0.66	0.09	0.98
C18:3 cis6 cis9 cis12	0.05	0.05	0.03	0.19
C18:3 cis9 cis12 cis15	0.71	1.06	0.24	0.05
C20:2 n-6	0.05	0.06	0.15	0.09
C20:3 n-6	0.04	0.03	0.16	0.45
C20:3 n-3	0.01	0.01	0.28	0.32
C20:4 n-6	0.04	0.05	0.19	0.09
C20:5 n-3	0.06	0.07	0.12	0.09
C22:2	0.10	0.14	0.24	0.06
C22:6 n-3	0.02	0.04	0.31	0.07
PUFA	3.15	3.85	0.13	0.05
n3	0.80	1.18	0.23	0.05
n6	2.22	2.45	0.14	0.51
n6/n3	2.82	2.10	0.25	0.27

* Individual standards not included in the mix. ^1^ SEM (standard error of mean).

**Table 2 foods-13-03320-t002:** Chemical composition, total polyphenols, and fatty acid profile of Caciocavallo cheeses produced from milk of cows in the CTR group (without ECO) and cows in the ECO group (with the integration of 7% of ECO in the concentrate).

Caciocavallo Cheese				
	Groups		SEM ^1^	*p*-Value
Item	CTR	ECO		
PH	-	-	-	-
SH°	-	-	-	-
Moisture (%)	37.74	39.96	0.01	0.04
Total proteins (%)	30.24	31.57	0.02	0.27
Casein (%)	-	-	-	-
Total lipids (%)	27.49	25.03	0.02	0.05
Lactose (%)	-	-	-	-
Total polyphenols (mg/kg)	110.91	232.85	0.18	0.03
Fatty acids (g/100 g FA)				
C4:0	2.73	2.65	0.07	0.27
C6:0	2.37	2.58	0.02	0.05
C8:0	1.80	1.74	0.02	0.27
C10:0	3.58	3.65	0.01	0.51
C11:0	0.03	0.06	0.13	0.04
C12:0	3.70	3.58	0.01	0.05
C13:0	0.09	0.11	0.05	0.04
C14:0	11.11	10.37	0.02	0.05
C15:0 iso *	0.25	0.23	0.03	0.10
C15:0 anteiso *	0.01	0.01	0.000	1.00
C15:0	1.00	1.01	0.01	0.66
C16:0	27.93	27.85	0.003	0.83
C17:0	0.55	0.46	0.06	0.05
C18:0	15.03	14.45	0.01	0.04
C20:0	0.94	1.04	0.03	0.51
C21:0	0.04	0.03	0.08	0.09
C22:0	0.05	0.07	0.07	0.03
C23:0	0.04	0.05	0.09	0.04
C24:0	0.04	0.04	0.07	0.19
SFA	71.29	69.98	0.005	0.05
C14:1	0.97	0.98	0.01	0.38
C15:1	0.21	0.28	0.07	0.05
C16:1	1.35	1.71	0.07	0.05
C16:1 trans *	0.01	0.01	0.000	1.00
C16:1 n-5	0.01	0.01	0.000	1.00
C17:1	0.59	0.52	0.06	0.51
C18:1 cis9	16.43	18.12	0.03	0.04
C18:1 trans9	1.67	0.98	0.12	0.05
C18:1 cis11 *	2.10	2.41	0.05	0.38
C18:1 trans11 *	0.01	0.02	0.15	0.11
C20:1 n-11	0.10	0.10	0.02	0.19
C22:1 n-9	0.01	0.01	0.000	1.00
C24:1 n-9	0.02	0.03	0.11	0.09
MUFA	23.48	25.19	0.02	0.03
C16:3 n-4 *	0.01	0.01	0.000	1.00
C18:2 cis9 cis12	0.75	0.93	0.05	0.05
C18:2 trans9 trans12	0.40	0.38	0.02	0.37
C18:3 cis6 cis9 cis12	0.27	0.31	0.04	0.05
C18:3 cis9 cis12 cis15	0.41	0.43	0.04	0.83
C20:2 n-6	0.05	0.05	0.05	0.19
C20:3 n-6	0.06	0.04	0.07	0.07
C20:3 n-3	0.01	0.01	0.000	1.00
C20:4 n-6	0.02	0.03	0.16	0.09
C20:5 n-3	0.06	0.07	0.08	0.09
C22:2	0.26	0.22	0.05	0.27
C22:6 n-3	0.01	0.02	0.15	0.11
PUFA	2.30	2.50	0.02	0.05
n3	0.48	0.53	0.03	0.12
n6	1.55	1.74	0.03	0.05
n6/n3	3.22	3.30	0.03	0.83

* Individual standards not included in the mix. ^1^ SEM (standard error of mean).

**Table 3 foods-13-03320-t003:** Microbial counts (expressed as mean log10 cfu/mL ± standard deviation) of milk samples. CTR group: milk from dairy cows fed concentrate without olive cake supplements. ECO group: milk from dairy cows fed concentrate with 7% enriched olive cake as 7% of a DM.

Microbial Groups	CTR	ECO	*p*-Value
*Enterococcus* spp.	3.04 ± 0.13	3.15 ± 0.15	0.34
*Enterobacteriaceae*	2.70 ± 0.35	2.43 ± 0.23	0.24
*S. aureus*	0.93 ± 1.60	0.95 ± 0.1.64	0.11
Total mesophilic bacteria	4.54 ± 0.25	4.42 ± 0.26	0.13
Yeasts and molds	2.57 ± 0.20	2.42 ± 0.10	0.84
*Lactococcus* spp.	3.11 ± 0.02	3.20 ± 0.02	0.41
*Thermophilic lactococci*	3.77 ± 0.04	5.09 ± 0.75	0.01
*Lactic acid bacteria*	4.55 ± 0.18	4.45 ± 0.08	0.41
*E. coli*	1.47 ± 1.29	0.67 ± 1.15	0.01
Total coliforms	1.80 ± 1.56	1.33 ± 1.15	0.41
*Salmonella* spp.	Absent	Absent	---
*L. monocytogenes*	Absent	Absent	---

**Table 4 foods-13-03320-t004:** Microbial counts (expressed as mean log10 cfu/g ± standard deviation) of cheese samples. CTR group: cheese made by dairy cows fed concentrate without olive cake supplements. ECO group: cheese made by dairy cows fed concentrate with 7% enriched olive cake as 7% of a DM.

Microbial Groups	CTR	ECO	*p*-Value
*Enterococcus* spp.	5.04 ± 0.19	5.15 ± 0.15	0.65
*Enterobacteriaceae*	2.54 ± 0.34	2.72 ± 0.17	0.33
Coagulase+ staphylococci	4.48 ± 0.01	5.29 ± 0.30	0.13
Total mesophilic bacteria	6.38 ± 0.11	6.23 ± 0.11	0.62
Yeasts and molds	6.57 ± 0.26	6.42 ± 0.17	0.84
*Lactococcus* spp.	6.91 ± 0.02	6.68 ± 1.42	0.35
*Thermophilic lactococci*	2.77 ± 0.04	5.29 ± 0.75	0.01
*Lactic acid bacteria*	7.34 ± 0.48	7.03 ± 0.28	0.41
*E. coli O157*	Absent	Absent	---
*E. coli*	Absent	Absent	---
*Salmonella* spp.	Absent	Absent	---
*L. monocytogenes*	Absent	Absent	---

**Table 5 foods-13-03320-t005:** Volatile composition (peak area % ± SD) of CTR and ECO concentrates and olive cake. CTR: concentrate without olive cake supplements. ECO: concentrate with 7% enriched olive cake as 7% of a DM.

Compounds	LRI	CTR	ECO	Olive Cake
Acids				
Acetic acid	1455	tr ^a^	tr ^a^	0.36 ± 0.02 ^b^
Propionic acid	1540	tr^a^	tr ^a^	0.23 ± 0.01 ^b^
2-Methyl-propanoic acid	1566	0.62 ± 0.04 ^b^	0.55 ± 0.04 ^b^	0.28 ± 0.02 ^a^
Butanoic acid	1629	tr ^a^	tr ^a^	0.94 ± 0.06 ^b^
2-Methyl-butanoic acid	1666	3.14 ± 0.12 ^c^	2.64 ± 0.22 ^b^	0.85 ± 0.05 ^a^
Hexanoic acid	1844	1.62 ± 0.18 ^a^	5.02 ± 0.41 ^b^	5.58 ± 0.41 ^b^
Heptanoic acid	1950	tr ^a^	0.77 ± 0.06 ^b^	0.85 ± 0.06 ^b^
Octanoic acid	2060	0.26 ± 0.03 ^a^	0.94 ± 0.07 ^b^	1.06 ± 0.08 ^b^
Nonanoic acid	2165	tr ^a^	0.71 ± 0.03 ^b^	0.74 ± 0.04 ^b^
Tetradecanoic acid	2694	0.90 ± 0.07 ^a^	0.87 ± 0.07 ^a^	0.39 ± 0.02 ^b^
Pentadecanoic acid	2799	0.26 ± 0.03 ^b^	0.26 ± 0.03 ^b^	tr ^a^
Hexadecanoic acid	2906	3.23 ± 0.16 ^a^	3.63 ± 0.18 ^b^	3.89 ± 0.31 ^b^
All		10.02 ± 0.63 ^a^	15.39 ± 1.11 ^b^	15.18 ± 1.08 ^c^
Aldehydes				
Pentanal	985	0.85 ± 0.05 ^a^	1.23 ± 0.07 ^b^	0.66 ± 0.04 ^a^
Hexanal	1085	9.79 ± 0.57 ^c^	8.45 ± 0.42 ^b^	4.42 ± 0.22 ^a^
Heptanal	1188	1.21 ± 0.08 ^a^	2.44 ± 0.17 ^b^	2.03 ± 0.14 ^b^
Octanal	1292	3.54 ± 0.14 ^a^	5.76 ± 0.23 ^b^	6.86 ± 0.27 ^c^
(*Z*)-2-Heptenal	1328	0.58 ± 0.03 ^a^	0.50 ± 0.02 ^a^	1.39 ± 0.07 ^b^
Nonanal	1395	3.71 ± 0.19 ^a^	10.31 ± 0.61 ^b^	24.04 ± 1.44 ^c^
(*E*)-2-Octenal	1431	0.96 ± 0.06 ^a^	0.92 ± 0.05 ^a^	0.80 ± 0.04 ^a^
Decanal	1499	0.37 ± 0.01 ^a^	0.42 ± 0.02 ^a^	1.25 ± 0.05 ^b^
Benzaldehyde	1527	0.24 ± 0.02 ^a^	0.55 ± 0.04 ^a^	2.01 ± 0.14 ^b^
(*E*)-2-Nonenal	1535	0.39 ± 0.02 ^b^	0.26 ± 0.02 ^b^	tr ^a^
(*E*)-2-Decenal	1645	0.29 ± 0.01 ^a^	0.93 ± 0.05 ^b^	0.80 ± 0.04 ^b^
(*Z*)-8-Undecenal	1750	tr ^a^	0.30 ± 0.02 ^b^	tr ^a^
All		21.95 ± 1.18 ^a^	32.07 ± 1.72 ^b^	44.27 ± 2.45 ^c^
Alcohols				
Ethanol	937	tr ^a^	tr ^a^	0.23 ± 0.01 ^b^
5-Ethyl-2-heptanol	1121	tr ^a^	tr ^a^	0.25 ± 0.02 ^b^
1-Butanol	1143	0.32 ± 0.02 ^b^	0.29 ± 0.01 ^b^	tr ^a^
2-Methyl-1-butanol	1158	0.26 ± 0.01 ^b^	tr ^a^	tr ^a^
3-Methyl-1-butanol	1205	0.39 ± 0.02 ^b^	0.27 ± 0.01 ^a^	0.27 ± 0.02 ^a^
1-Pentanol	1247	6.27 ± 0.34 ^c^	4.75 ± 0.19 ^b^	tr ^a^
Heptan-2-ol	1315	0.35 ± 0.02 ^b^	tr ^a^	tr ^a^
3-Methyl-1-pentanol	1351	25.90 ± 1.18 ^c^	13.40 ± 0.40 ^b^	2.06 ± 0.06 ^a^
1-Octen-3-ol	1444	5.27 ± 0.26 ^c^	4.04 ± 0.18 ^b^	2.08 ± 0.04 ^a^
1-Heptanol	1450	7.57 ± 0.45 ^c^	5.77 ± 0.28 ^b^	0.29 ± 0.01 ^a^
6-Methyl-5-hepten-2-ol	1457	0.38 ± 0.04 ^b^	0.34 ± 0.03 ^b^	tr ^a^
2,4-Dimethyl-cyclohexanol	1478	0.31 ± 0.02 ^b^	tr ^a^	tr ^a^
2-Ethyl-1-hexanol	1484	0.26 ± 0.01 ^b^	tr ^a^	0.28 ± 0.02 ^b^
1-Octanol	1553	7.21 ± 0.10 ^c^	6.78 ± 0.06 ^b^	0.98 ± 0.01 ^a^
(*Z*)-2-Octen-1-ol	1612	tr ^a^	0.23 ± 0.01 ^b^	0.26 ± 0.01 ^b^
1-Nonanol	1656	0.87 ± 0.04 ^b^	1.17 ± 0.06 ^c^	tr ^a^
Guaiacol	1859	tr ^a^	tr ^a^	0.29 ± 0.02 ^b^
Benzyl alcohol	1875	tr ^a^	0.47 ± 0.04 ^b^	0.89 ± 0.07 ^b^
Phenethyl alcohol	1909	tr ^a^	0.58 ± 0.05 ^b^	1.10 ± 0.09 ^c^
Creosol	1954	tr ^a^	tr ^a^	3.30 ± 0.26 ^b^
4-Ethyl-phenol	2176	tr ^a^	0.64 ± 0.05 ^b^	6.04 ± 0.48 ^c^
All		55.37 ± 2.51 ^c^	38.74 ± 1.37 ^b^	18.33 ± 1.12 ^a^
Ketones				
Heptan-2-one	1185	1.21 ± 0.09 ^b^	0.90 ± 0.07 ^a^	0.78 ± 0.05 ^a^
2-Octanone	1287	0.47 ± 0.03 ^a^	0.43 ± 0.02 ^a^	0.72 ± 0.06 ^b^
1-Octen-3-one	1304	tr ^a^	tr ^a^	0.25 ± 0.02 ^b^
6-Methyl-5-hepten-2-one	1336	2.27 ± 0.18 ^b^	1.67 ± 0.11 ^a^	2.12 ± 0.09 ^b^
4-Ethyl-cyclohexanone	1344	0.51 ± 0.03 ^b^	tr ^a^	tr ^a^
2-Nonanone	1389	0.60 ± 0.04 ^a^	0.66 ± 0.03 ^a^	1.79 ± 0.13 ^b^
Oct-3-en-2-one	1409	1.96 ± 0.14 ^a^	3.07 ± 0.31 ^b^	3.40 ± 0.45 ^b^
6-Methoxy-2-hexanone	1421	0.27 ± 0.03 ^b^	tr ^a^	tr ^a^
2-Decanone	1492	0.29 ± 0.04 ^a^	0.45 ± 0.05 ^b^	0.48 ± 0.05 ^b^
3,5-Octadien-2-one	1520	tr ^a^	0.52 ± 0.03 ^b^	0.31 ± 0.01 ^b^
(*E,E*)-3,5-Octadien-2-one	1572	0.48 ± 0.04 ^b^	0.41 ± 0.03 ^b^	tr ^a^
6-Methyl-3,5-heptadiene-2-one	1594	tr ^a^	0.40 ± 0.02 ^b^	1.02 ± 0.05 ^c^
Acetophenone	1652	tr ^a^	tr ^a^	0.35 ± 0.02 ^b^
Nerylacetone	1851	tr ^a^	tr ^a^	0.29 ± 0.01 ^b^
All		8.06 ± 0.62 ^a^	8.51 ± 0.67 ^a^	11.51 ± 0.94 ^b^
Esters				
Methyl acetate	833	tr ^a^	tr ^a^	0.68 ± 0.04 ^b^
Ethyl acetate	893	tr ^a^	tr ^a^	0.55 ± 0.03 ^b^
Methyl propionate	912	tr ^a^	tr ^a^	0.38 ± 0.03 ^b^
Ethyl propanoate	961	tr ^a^	tr ^a^	0.29 ± 0.02 ^b^
Methyl butyrate	991	tr ^a^	tr ^a^	0.25 ± 0.02 ^b^
Ethyl butanoate	1038	tr ^a^	tr ^a^	0.41 ± 0.03 ^b^
Methyl nonanoate	1489	tr ^a^	tr ^a^	0.71 ± 0.05 ^b^
Pentyl hexanoate	1511	0.24 ± 0.02 ^a^	0.50 ± 0.04 ^b^	0.51 ± 0.04 ^b^
Hexyl hexanoate	1607	tr ^a^	0.66 ± 0.05 ^b^	Tr ^a^
All		0.24 ± 0.02 ^a^	1.17 ± 0.09 ^b^	3.78 ± 0.26 ^c^
Lactones				
γ-Octalactone	1916	0.36 ± 0.02 ^b^	0.76 ± 0.05 ^c^	tr ^a^
γ-Nonalactone	2028	1.09 ± 0.07 ^a^	1.43 ± 0.12 ^b^	6.04 ± 0.47 ^c^
All		1.45 ± 0.09 ^a^	2.19 ± 0.17 ^b^	6.04 ± 0.47 ^c^
Oxides				
cis-Linalool oxide	1439	0.95 ± 0.08 ^c^	0.58 ± 0.03 ^b^	tr ^a^
trans-Linalool oxide (furanoid)	1467	0.44 ± 0.02 ^c^	0.31 ± 0.01 ^b^	tr ^a^
All		1.39 ± 0.10 ^c^	0.89 ± 0.04 ^b^	tr ^a^
Furans				
2-Pentyl-furan	1230	1.52 ± 0.12 ^c^	1.06 ± 0.07 ^b^	0.89 ± 0.05 ^a^
All		1.52 ± 0.12 ^c^	1.06 ± 0.07 ^b^	0.89 ± 0.05 ^a^

LRI = Linear retention index. tr = traces. Different superscript lowercase letters in the same row indicate significant differences at *p* ≤ 0.05 among samples by Duncan’s multiple range test.

**Table 6 foods-13-03320-t006:** Volatile constituents (peak area % ± SD) of CTR and ECO cheese and milk samples. CTR: milk and cheese made by dairy cows fed concentrate without olive cake supplements. ECO: milk and cheese made by dairy cows fed concentrate with 7% enriched olive cake as 7% of a DM.

		Cheese			Milk		
Compounds	LRI	CTR	ECO		CTR	ECO	
Acids							
Acetic acid	1460	0.90 ± 0.06 ^b^	0.33 ± 0.02 ^a^	**	2.19 ± 0.15 ^b^	0.13 ± 0.01 ^a^	***
Propanoic acid	1543	tr ^a^	0.46 ± 0.03 ^b^	***	-	-	ns
Butanoic acid	1631	3.24 ± 0.16 ^b^	1.14 ± 0.07 ^a^	**	0.51 ± 0.04 ^a^	0.93 ± 0.03 ^b^	*
Hexanoic acid	1740	9.09 ± 0.63 ^b^	3.90 ± 0.31 ^a^	*	1.07 ± 0.05 ^a^	1.05 ± 0.04 ^a^	ns
Heptanoic acid	1954	0.22 ± 0.01 ^a^	0.16 ± 0.01 ^a^	*	0.37 ± 0.02 ^b^	Tr ^a^	*
Octanoic acid	2062	3.79 ± 0.18 ^b^	2.88 ± 0.14 ^a^	*	1.21 ± 0.06 ^a^	1.65 ± 0.08 ^b^	ns
Nonanoic acid	2170	2.02 ± 0.10 ^a^	1.96 ± 0.08 ^a^	ns	3.49 ± 0.21 ^b^	1.09 ± 0.06 ^a^	**
(E)-2-Octenoic acid	2184	tr ^a^	0.36 ± 0.02 ^b^	**	-	-	ns
Decanoic acid	2276	2.24 ± 0.11 ^b^	1.87 ± 0.07 ^a^	*	5.21 ± 0.36 ^b^	4.00 ± 0.28 ^a^	ns
(E)-9-Decenoic acid	2332	-	-	ns	0.08 ± 0.01 ^a^	tr ^a^	ns
Undecanoic acid	2379	0.27 ± 0.02 ^a^	0.23 ± 0.02 ^a^	ns	0.26 ± 0.02 ^a^	0.29 ± 0.02 ^a^	ns
(E)-2-Decenoic acid	2399	0.15 ± 0.01 ^a^	0.37 ± 0.03 ^a^	*	0.24 ± 0.02 ^a^	0.32 ± 0.03 ^a^	ns
Benzoic acid	2430	-	-	ns	0.32 ± 0.01 ^a^	0.29 ± 0.02 ^a^	ns
Dodecanoic acid	2488	1.70 ± 0.12 ^a^	2.03 ± 0.16 ^a^	ns	2.32 ± 0.13 ^a^	3.25 ± 0.23 ^b^	ns
Tridecanoic acid	2587	tr ^a^	0.23 ± 0.01 ^a^	*	0.34 ± 0.02 ^a^	0.54 ± 0.04 ^b^	ns
Tetradecanoic acid	2698	6.27 ± 0.25 ^a^	11.57 ± 0.81 ^b^	*	22.69 ± 1.34 ^a^	25.67 ± 1.26 ^b^	ns
Pentadecanoic acid	2799	2.50 ± 0.15 ^a^	3.34 ± 0.23 ^b^	ns	3.13 ± 0.10 ^a^	3.44 ± 0.08 ^b^	ns
Hexadecanoic acid	2810	24.42 ± 1.22 ^a^	41.36 ± 1.65 ^b^	*	43.50 ± 2.26 ^a^	48.72 ± 2.59 ^b^	ns
All		56.81 ± 3.02 ^a^	72.19 ± 3.66 ^b^	*	86.93 ± 4.80 ^a^	91.37 ± 4.77 ^b^	ns
Short-Medium		21.92 ± 1.28 ^b^	13.66 ± 0.80 ^a^	*	14.63 ± 0.94 ^b^	9.46 ± 0.55 ^a^	*
Long		34.89 ± 1.74 ^a^	58.53 ± 2.86 ^b^	*	71.98 ± 3.86 ^a^	81.91 ± 4.22 ^b^	*
Ratio S-M/L		0.63 ± 0.74 ^a^	0.23 ± 0.28 ^a^	*	0.20 ± 0.24 ^b^	0.11 ± 0.13 ^a^	*
Alcohols							
Ethanol	938	0.74 ± 0.05 ^a^	4.15 ± 0.54 ^b^	***	0.16 ± 0.01 ^a^	tr ^a^	*
2-Methyl-propanol	1096	0.27 ± 0.02 ^a^	0.13 ± 0.01 ^a^	*	-	-	ns
Isoamyl alcohol	1206	1.73 ± 0.12 ^a^	5.39 ± 0.85 ^b^	*	-	-	ns
3-Methyl-butanol	1212	-	-	ns	0.04 ± 0.01 ^a^	tr ^a^	ns
Pentanol	1250	-	-	ns	0.15 ± 0.01 ^a^	0.20 ± 0.02 ^a^	ns
3-Methyl-2-buten-1-ol	1319	tr ^a^	0.07 ± 0.01 ^a^	*	-	-	ns
Hexanol	1350	0.08 ± 0.01 ^a^	tr ^a^	*	0.08 ± 0.01 ^a^	tr ^a^	ns
2-Ethyl-hexanol	1482	0.61 ± 0.04 ^a^	0.42 ± 0.03 ^a^	ns	0.61 ± 0.04 ^b^	0.42 ± 0.03 ^a^	ns
Octanol	1550	0.13 ± 0.01 ^a^	0.10 ± 0.01 ^a^	ns	0.13 ± 0.01 ^a^	0.10 ± 0.01 ^a^	ns
Dodecanol	1970	tr ^a^	0.72 ± 0.05 ^b^	*	-	-	ns
Tetradecanol	2174	tr ^a^	0.29 ± 0.03 ^a^	*	-	-	ns
All		3.56 ± 0.25 ^a^	11.27 ± 1.53 ^b^	**	1.17 ± 0.09 ^b^	0.72 ± 0.06 ^a^	ns
Aldehydes							
Pentanal	984	-	-	ns	0.31 ± 0.02 ^b^	0.21 ± 0.01 ^a^	ns
Hexanal	1085	1.48 ± 0.07 ^b^	1.06 ± 0.04 ^a^	ns	1.58 ± 0.13 ^b^	0.67 ± 0.04 ^a^	*
Heptanal	1189	0.82 ± 0.04 ^b^	0.32 ± 0.02 ^a^	*	-	-	ns
Octanal	1282	-	-	ns	0.35 ± 0.02 ^b^	tr ^a^	*
Nonanal	1394	2.17 ± 0.15 ^a^	1.99 ± 0.11 ^a^	ns	1.26 ± 0.11 ^a^	1.30 ± 0.12 ^b^	ns
Decanal	1491	-	-	ns	0.78 ± 0.06 ^b^	tr ^a^	*
Undecanal	1598	-	-	ns	0.47 ± 0.03 ^b^	tr ^a^	*
All		4.47 ± 0.26 ^b^	3.37 ± 0.17 ^a^	ns	4.75 ± 0.37 ^b^	2.18 ± 0.17 ^a^	*
Ketones							
2-Butanone	910	0.84 ± 0.06 ^a^	0.52 ± 0.04 ^a^	ns	1.67 ± 0.11 ^b^	1.47 ± 0.09 ^a^	ns
2,3-Butanedione	985	8.76 ± 0.35 ^b^	1.61 ± 0.08 ^a^	***	-	-	ns
2,3-Pentanedione	1062	0.24 ± 0.02 ^a^	0.12 ± 0.01 ^a^	*	-	-	ns
3-Heptanone	1155	0.08 ± 0.01 ^a^	tr ^a^	*	-	-	ns
2-Heptanone	1185	10.01 ± 0.48 ^b^	1.54 ± 0.12 ^a^	***	0.56 ± 0.04 ^a^	0.70 ± 0.05 ^b^	ns
Acetoin	1294	9.82 ± 0.59 ^b^	5.63 ± 0.22 ^a^	*	-	-	ns
2,5-Octanedione	1316	-	-	ns	0.11 ± 0.01 ^b^	tr ^a^	*
6-Methyl-5-hepten-2-one	1338	0.12 ± 0.01 ^a^	0.10 ± 0.01 ^a^	ns	-	-	ns
2-Nonanone	1389	0.48 ± 0.03 ^a^	0.11 ± 0.01 ^a^	**	0.16 ± 0.01 ^b^	0.09 ± 0.07 ^a^	*
All		30.35 ± 1.55 ^b^	9.63 ± 0.49 ^a^	***	2.50 ± 0.17 ^b^	2.26 ± 0.21 ^a^	ns
Esters							
Ethyl acetate	896	0.26 ± 0.03 ^a^	0.66 ± 0.05 ^a^	*	-	-	ns
Ethyl propanoate	962	0.04 ± 0.01 ^a^	0.22 ± 0.01 ^a^	***	-	-	ns
Ethyl butanoate	1039	0.31 ± 0.02 ^a^	0.63 ± 0.04 ^a^	*	0.13 ± 0.01 ^b^	tr ^a^	*
Isoamyl acetate	1122	0.32 ± 0.03 ^a^	0.09 ± 0.01 ^a^	**	-	-	ns
Methyl propyl butanoate	1157	0.12 ± 0.01 ^a^	tr ^a^	*	-	-	ns
Butyl butanoate	1218	0.15 ± 0.01 ^a^	tr ^a^	*	-	-	ns
Ethyl hexanoate	1232	tr ^a^	0.09 ± 0.01 ^a^	*	0.05 ± 0.01 ^a^	0.09 ± 0.01 ^a^	ns
Isopentyl butanoate	1264	0.32 ± 0.02 ^a^	tr ^a^	**	-	-	ns
Methyl hexadecanoate	2216	-	-	ns	0.23 ± 0.01 ^a^	0.64 ± 0.04 ^b^	*
All		1.52 ± 0.13 ^a^	1.69 ± 0.12 ^a^	ns	0.41 ± 0.03 ^a^	0.73 ± 0.05 ^a^	*
Aromatic hydrocarbons							
Ethylbenzene	1128	0.18 ± 0.01 ^b^	0.05 ± 0.01 ^a^	**	tr ^a^	tr ^a^	ns
Styrene	1262	0.61 ± 0.04 ^b^	0.22 ± 0.02 ^a^	**	tr ^a^	tr ^a^	ns
All		0.79 ± 0.05 ^b^	0.27 ± 0.03 ^a^	**	tr ^a^	tr ^a^	ns
Lactones							
δ-Decalactone	2197	tr ^a^	0.29 ± 0.03 ^b^	*	-	0.29 ± 0.02 ^b^	*
All		tr ^a^	0.29 ± 0.03 ^b^	*	-	0.29 ± 0.02 ^b^	*
Terpenes							
α-Pinene	1024	tr ^a^	tr ^a^	ns	0.32 ± 0.02 ^a^	0.55 ± 0.04 ^b^	*
δ-3-Carene	1146	0.02 ± 0.01 ^a^	tr ^a^	ns	-	-	ns
Myrcene	1160	0.16 ± 0.01 ^b^	tr ^a^	*	-	-	ns
α-Terpinene	1180	0.04 ± 0.01 ^a^	tr ^a^	ns	-	-	ns
Limonene	1201	1.21 ± 0.01 ^a^	1.38 ± 0.08 ^a^	ns	0.36 ± 0.03 ^b^	tr ^a^	*
Eucalyptol	1211	0.64 ± 0.05 ^b^	0.13 ± 0.01 ^a^	**	0.17 ± 0.01 ^a^	0.10 ± 0.01 ^a^	ns
p-Cymene	1272	0.31 ± 0.03 ^a^	0.33 ± 0.02 ^a^	ns	0.72 ± 0.06 ^b^	tr ^a^	*
All		2.38 ± 0.12 ^b^	1.84 ± 0.11 ^a^	ns	1.57 ± 0.12 ^b^	0.65 ± 0.05 ^a^	*

LRI = Linear retention index. tr = traces. - = not detected. Different superscript lowercase letters in the same row indicate significant differences at *p* < 0.05 among samples by Duncan’s multiple range test; Statistically significant differences among CTR and ECO cheese and milk at *p* < 0.001 (***), *p* < 0.01 (**) or *p* ≤ 0.05 (*), ns = not statistically significant (*p* > 0.05).

## Data Availability

The original contributions presented in the study are included in the article/Appendix A, further inquiries can be directed to the corresponding authors.

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
