# Peer review of "Feeding Cows with Olive Cake Enriched in Polyphenols Improves the Sustainability and Enhances the Nutritional and Organoleptic Features of Fresh Caciocavallo Cheese"

_foods, 2024, doi:10.3390/foods13203320_

Round 1
Reviewer 1 Report
Comments and Suggestions for Authors
The aim of presented study was to evaluate the nutritional value, microbiological profiles, sensory features, and volatile aroma compounds of Caciocavallo Ragusano produced with milk obtained from cows fed with olive cake, enriched in polyphenols.
Therefore, I suggest changing the title of the work e.g. to: „The use in cow nutrition of olive cake enriched in polyphenols and its influence on the nutritional value,
microbiological profiles, sensory features, and volatile aroma compounds
of Caciocavallo Ragusano cheese”.
In the chapter Materials and Methods, I suggest briefly describing:
- The management of the cows and the nutritional characteristics of the enriched olive cake (Line 74-75),
- A flowchart of enriched OC production (Line 80),
- A flowchart od Caciocavallo Ragusano production (Line 88),
and not just refer to literature.
The abbreviation OC should be entered after the first use of the phrase olive cake (Line 45).
In the Line 77 the abbreviations OC and DM appear without explanation.
The scope of the analyzes is very extensive, and the research and statistical methodology used is correct.
In the Line 197 it should be: „… at p ≤ 0.05”. The same in the Lines 221, 310, 391.
Uniform abbreviations should be used in the table heads, e.g. in table 5 there is Control instead of CTR, and in table 6 there is Control and Biotrack instead of CTR and ECO
In the table head 4 there is P-value instead p-value.
The conclusions are too general. There are no conclusions from microbiological analyses, including PCR and volatile compounds. I suggest extending the conclusions to the statements contained in the summary of the work.
Author Response
The aim of presented study was to evaluate the nutritional value, microbiological profiles, sensory features, and volatile aroma compounds of Caciocavallo Ragusano produced with milk obtained from cows fed with olive cake, enriched in polyphenols.
Therefore, I suggest changing the title of the work e.g. to: „The use in cow nutrition of olive cake enriched in polyphenols and its influence on the nutritional value,
microbiological profiles, sensory features, and volatile aroma compounds
of Caciocavallo Ragusano cheese”.
AU: We would like to thank the Reviewer for the suggestion. We provided a new title accordingly to the reviewer’s suggestion.
In the chapter Materials and Methods, I suggest briefly describing:
- The management of the cows and the nutritional characteristics of the enriched olive cake (Line 74-75),
- A flowchart of enriched OC production (Line 80),
- A flowchart od Caciocavallo Ragusano production (Line 88),
and not just refer to literature.
AU: Thank you for the comment. The management and nutritional characteristic of enriched olive cake were reported respectively in line 79-86 and in Table S1 of the supplementary file. The flowchart of enriched olive cake was reported in Figure 1, while the flowchart of Caciocavallo Ragusano production in reported in Figure S1 of the supplementary file.
The abbreviation OC should be entered after the first use of the phrase olive cake (Line 45).
AU: Added, thank you.
In the Line 77 the abbreviations OC and DM appear without explanation.
AU: Thanks, the abbreviation DM was clarified in line 88.
The scope of the analyzes is very extensive, and the research and statistical methodology used is correct.
In the Line 197 it should be: „… at p ≤ 0.05”. The same in the Lines 221, 310, 391.
AU: Modified, thanks.
Uniform abbreviations should be used in the table heads, e.g. in table 5 there is Control instead of CTR, and in table 6 there is Control and Biotrack instead of CTR and ECO
AU: We modified the abbreviation, thank you.
In the table head 4 there is P-value instead p-value.
AU: Modified, thanks.
The conclusions are too general. There are no conclusions from microbiological analyses, including PCR and volatile compounds. I suggest extending the conclusions to the statements contained in the summary of the work.
AU: Thank you for the comment. The conclusion was extended with the microbiological analyses and the volatile compound, as suggested.
Reviewer 2 Report
Comments and Suggestions for Authors
The manuscript is well-written and easy to follow. But there are some minor suggestions:
Page 2, line 77: OC - When some abbreviation appears in the text for the first time, it must be clearly explained.
Page 3, lines 97-98: All AOAC methods need to be listed in the References list.
Page 3, line 133: DGGE - Please, write it in full-term form.
Page 4, lines 161-162: ISO 8586-1:1993 needs to be added to the Reference list.
Page 4, line 186: SPSS needs to be added to the Reference list.
Page 5, lines 195-196: Statistica software (version 10.0 for Windows,
TIBCO Software, Palo Alto, CA, USA needs to be added to the Reference list.
Page 5, line 211: The authors write fat content, but in Table 2 this parameter is expressed as Total lipids. Please, uniform this term in the Table and text.
For all Tables and Figures: The abbreviations CTR and ECO need to be explained in the footnote. Each table needs to be self-explanatory without referring to the main text.
Page 7, lines 216-219: Please, emphasize that you had similar trends of fatty acid composition in milk and cheeses. What was the same, and what was different considering milk and cheese?
Page 8, line 236: Please correct Vargàs-Bello-Pérez et al. [36] to Tzamaloukas et al. [36].
Page 8, line 240: Autho's results must be expressed in the past tense.
Page 8, lines 264-269: analysis of the correlations I couldn't follow in Figure 1. The resolution of the Figure 1 is bad.
Page 8, line 265: Please instead "for" write "between".
Page 9, line 284: Commission Regulation (EC) No 2073/2005 needs to be added to the Reference list.
Table 5. please uniform terms. Please do not write "Control" write CTR.
Table 6. Please uniform terms. Please do not write "Control" and "Biotrak" but write CTR and ECO.
Figure 4. Please, explain in short what 1 means, and what means 9 (unit). Each figure needs to be self-explanatory without reference to the main text.

Author Response
The manuscript is well-written and easy to follow. But there are some minor suggestions:
Page 2, line 77: OC - When some abbreviation appears in the text for the first time, it must be clearly explained.
AU: Thanks, the therm olive cake was clarified with the abbreviation.
Page 3, lines 97-98: All AOAC methods need to be listed in the References list.
AU: Added, thanks.
Page 3, line 133: DGGE - Please, write it in full-term form.
AU: The abbreviation was clarified, thanks.
Page 4, lines 161-162: ISO 8586-1:1993 needs to be added to the Reference list.
AU: The reference was added, thanks.
Page 4, line 186: SPSS needs to be added to the Reference list.
AU: The reference was added, thanks.
Page 5, lines 195-196: Statistica software (version 10.0 for Windows,
TIBCO Software, Palo Alto, CA, USA needs to be added to the Reference list.
AU: The reference was added, thanks.
Page 5, line 211: The authors write fat content, but in Table 2 this parameter is expressed as Total lipids. Please, uniform this term in the Table and text.
AU: Done, we reported in the text total lipids, as in the table. Thanks.
For all Tables and Figures: The abbreviations CTR and ECO need to be explained in the footnote. Each table needs to be self-explanatory without referring to the main text.
AU: Thanks for the suggestion. Instead of describe the abbreviations in the footnote, we explained them in the legend at the beginning of the table.
Page 7, lines 216-219: Please, emphasize that you had similar trends of fatty acid composition in milk and cheeses. What was the same, and what was different considering milk and cheese?
AU: Thanks for the suggestion. We revised accordingly.
Page 8, line 236: Please correct Vargàs-Bello-Pérez et al. [36] to Tzamaloukas et al. [36].
AU: Sorry for the mistake. Corrected.
Page 8, line 240: Autho's results must be expressed in the past tense.
AU: Changed in “resulted”, as suggested.
Page 8, lines 264-269: analysis of the correlations I couldn't follow in Figure 1. The resolution of the Figure 1 is bad.
AU: Thanks. We provided the new figure with greater resolution.
Page 8, line 265: Please instead "for" write "between".
AU: Changed. Thanks.
Page 9, line 284: Commission Regulation (EC) No 2073/2005 needs to be added to the Reference list.
AU: Added to the Reference list. Thanks.
Table 5. please uniform terms. Please do not write "Control" write CTR.
AU: Changed.
Table 6. Please uniform terms. Please do not write "Control" and "Biotrak" but write CTR and ECO.
AU: Changed.
Figure 4. Please, explain in short what 1 means, and what means 9 (unit). Each figure needs to be self-explanatory without reference to the main text.
AU: Thanks. Modified